# Analysis of Conductance Probes for Two-Phase Flow and Holdup Applications

**DOI:** 10.3390/s20247042

**Published:** 2020-12-09

**Authors:** José-Luis Muñoz-Cobo, Yago Rivera, Cesar Berna, Alberto Escrivá

**Affiliations:** Instituto de Ingeniería Energética, Universitat Politécnica de Valencia, 46022 Valencia, Spain; yaridu@upv.es (Y.R.); ceberes@iie.upv.es (C.B.); aescriva@iqn.upv.es (A.E.)

**Keywords:** conductance probes, two-phase flow sensors, liquid fraction determination from relative conductance, two-phase flow tomography

## Abstract

In this paper we perform an analysis of the conductance probes used in two-phase flow applications especially for two-phase flow tomography of annular flow, to measure the waves produced in the interface with different boundary conditions without perturbing the flow, and in addition we examine the holdup applications as measuring the average void fraction in a given region. The method used to obtain the detector conductance between the electrodes is to solve analytically the generalized Laplace equation in 3D with the boundary conditions of the problem, and then to obtain the average potential difference between the detector electrodes. Then, dividing the current intensity circulating between the emitter and the receiver electrodes by the average potential difference yields the probe conductance, which depends on the geometric and physical characteristics of the measured system and the probe. This conductance is then non-dimensionalized by dividing by the conductance of the pipe full of water. In this way a set of analytical expression have been obtained for the conductance of two-plate sensors with different geometries and locations. We have performed an exhaustive comparison of the results obtained using the equations deduced in this paper with the experimental data from several authors in different cases with very good agreement. In some cases when the distribution of bubbles is not homogeneous, we have explored the different alternatives of the effective medium theory (EMT) in terms of the self-consistent EMT and the non-consistent EMT.

## 1. Introduction

Two-phase flow appears in a wide variety of applications in the chemical and petrochemical industries, energy industries like nuclear or concentrated solar power, civil engineering and so on. Different liquid and vapor flow patterns are found in the applications, which denote different topologies or configurations of the liquid and vapor distribution inside the pipe, channel or vessel containing the two-phase flow [1]. Notice that each flow pattern corresponds also to a characteristic distribution of the interfaces between the fluid phases. Each flow pattern depends on a set of conditions such as: pressure, superficial velocities of the liquid and vapor phases, temperature of each phase, heat flux through the walls, and geometry [2]. One of the more important flow regimes found in the applications is annular flow, which is characterized by a thin liquid film flowing adjacent to the walls of the pipe while a gas flow that usually transport entrained drops flows through the central part of the pipe. Usually, waves of different kinds are formed at the interface of this liquid film with the vapor or steam [3,4,5]. In addition, in many engineering applications the determination of the liquid fraction in two- and three-phase systems such as some fluidized beds is very important [6,7,8].

The main goal of this paper is to study the different types of conductance probes analytically and to compare these analytical results with numerical results and experiments of different authors to apply them to two-phase flow tomography and hold-up applications. The advantage of developing analytical models in 3D is that they allow a good design of the conductance probe selecting the most convenient size of the electrodes, distance between them, and type such as ring or plate. However, the developed models must be checked experimentally to ensure their ability to performs good experimental predictions. Once their prediction capability has been checked, they can be used to compare different conductance probes with different geometries and characteristics. 

The main advantage of the flush-mounted conductance probes is that the two-phase flow, assuming perfect device manufacturing, is not perturbed by the probe. This issue is important for the analysis of annular two-phase flows, especially when the film thickness is very thin. In this case, the sensor design should not perturb the waves produced at the film surface, since a small disturbance would introduce appreciable percent errors in the experimental measurements. There exist different types of conductance probes that have been designed for different applications in the past: the first one is the ring electrode probe formed by two ring shape electrodes, which are mounted along the circumference of the pipe perpendicular to the flow direction; this type of electrode has been studied by Fossa [8], and Tsochatzidis et al. [7]. There exist also four-ring electrode probes that have been used by Lina and Yingwei [9] to measure the water fraction in oil-water annular two-phase flow where the oil circulates through the core region of the pipe and the water flows close to the wall forming an annulus. Coney [10] measured the thickness of a rapidly varying wavy film by using a probe consisting in two parallel rectangular electrodes of length l≫2a, being 2*a* the distance between the electrodes. These electrodes are surface mounted in the pipe to not perturb the flow and are parallel to the flow direction. Also, Coney developed the three electrode probes segmenting the receiving electrode in two parts and measuring the ratio of the intensities flowing from the emitter electrode to the two receiver electrodes; this design has the advantage of compensating for changes in conductivity due to temperature. Recently Lee et al. [11] used the three-electrode probe, based on the ratio of the currents, to measure the film thickness under temperature-varying conditions because of the ratio of intensities is independent of the fluid conductivity changes with the temperature. In addition, Fossa [8] also performed measurements with two plate electrodes of 3mm diameter, flush mounted with a separation of 9 mm in the pipe axial direction. Finally, Ko et al. [12] and Lee et al. [13] designed recently improved electrical conductance sensors to perform void fraction measurements. Normally a high frequency alternating current (AC) is applied to the emitter electrode to avoid high gradients of ions and redox electrochemical reactions in the electrodes, which will degrade them.

The main novelty of this paper is that we have developed analytical expressions for the absolute and the relative conductance of two-plate conductance probes when the two sensors are mounted parallel to the flow direction or orthogonal to the flow direction in a 3D geometry with the goal of improving the results given by the Coney expression [10]. In addition, we computed the electric potential distribution generated by the two-plate electrode sensors in the film annulus. Moreover, other goal of the paper is to validate the new analytical expressions with experimental data and numerical calculations from different authors to know their limitations and potential range of applications. 

There exist also numerical methods for sensor design by solving numerically the generalized Laplace equation using the finite element method (FEM) as shown by Lee et al. [13], and Ko et al. [12]. These authors use the commercial program COMSOL Multiphysics to perform numerical calculations.

The paper has been organized as follows, first in Section 2.1, we deduce the expression for the electric potential and the relative conductance for ring conductance probes. In Section 2.2 we deduce the expressions for the electric potential and the relative conductance for two-plate conductance probes in two cases when the plate electrodes are located in the direction of the flow and when they are mounted orthogonal to the flow along the inner circumferential direction of the pipe. In Section 3.1 we perform the comparison of the expressions deduced in this paper for two-plate conductance probes with the experiments of Fossa [8], using two-plate electrodes along the flow direction. In Section 3.2 we perform a comparison of the variations of the relative conductance with the fraction of liquid (hold-up) for homogeneous bubbly flow with the experimental results of Fossa [8], and we discuss the influence of the effective-conductivity calculation on the results. In addition, in Section 3.3 we compare the results obtained using the expression deduced in this paper with the experimental results obtained by Coney [10] for the relative conductance, when changing the liquid fraction and using electrodes of different lengths that are parallel to the flow direction. Additionally, also in this section, we compare the results obtained with Coney formulas for two electrode probes of infinite electrode length with the results of our analytical expression when the length of the electrodes becomes large. In addition, in Section 3.4 we compare the experimental results obtained by Ko et al. [12] and Lee et al. [13] using static experiments in annular flow 4. Finally, in Section 4 we discuss the main results and findings of this paper.

## 2. Conductance Probes

### 2.1. The Two-Ring Conductance Probe

To compute the electric field between the electrodes when a high-frequency electric field is applied to the emitting electrode, one must consider the displacement current density in addition to the normal current. In this way, applying the operator (∇.) to the Henry law in the frequency domain yields [13,14]:(1)$$\nabla .\left(\nabla \times H\right)=\nabla .\left(j+j_{D}\right)=\nabla .\left(\sigma E+i\omega D\right)=-\nabla .\left(\sigma +i\omega \varepsilon \right).\nabla \phi =0$$
where $j,\;\mathrm{and}\;j_{D}$, are the current and displacement current densities respectively, σ is the electric conductivity, ***E*** is the intensity of the electric field between the electrodes, ω is the angular frequency, D is the electric displacement; ε is the dielectric constant and finally ϕ the electric field potential. It is necessary to avoid electrode polarization that will degrade the electrodes and produce capacitance effects. The way to achieve this is by applying a high-frequency alternate-current voltage source to the emitter electrode. The term of Equation (1), which contains ω is normally very small compared with the term containing the electric conductivity because of the electric permittivity is very small so we can neglect it and write Equation (1) as follows [13]:(2)∇.σ∇ϕ=0

In the interface between the conducting fluid film and the gas phase or the dielectric when we have a dielectric in the central part of the pipe, the continuity condition of the current density at both sides of the interface holds, i.e., we can set the following condition using cylindrical coordinates (r,θ,z) for the pipe geometry at the interface of radius $R_{in}$:(3)$${\left[-\sigma_{w}\frac{\partial \phi }{\partial r}\right]}_{r=R_{in}}={\left[-\sigma_{g}\frac{\partial \phi }{\partial r}\right]}_{r=R_{in}}$$
being $\sigma_{w}$ and $\sigma_{g}$ the conductivities of the water and gas phases, respectively. For the case in which we have a dielectric of conductivity $\sigma_{diel}$, and radius $R_{in}$ as displayed in Figure 1b we must substitute $\sigma_{g}$, by $\sigma_{diel}$ in Equation (3).

In addition, we must set the boundary conditions at the pipe walls. In this case, the boundary conditions depend on the type of the electrodes that are being used, the number of electrodes, and the current density or the electric potential in these electrodes. If we have two ring electrodes flush mounted as displayed in Figure 1a then the boundary conditions at the pipe inner surface of radius R, are:(4)$${\left[-\sigma_{w}\frac{\partial \phi }{\partial r}\right]}_{r=R}=j,\;\mathrm{for}\left(D_{e}-s_{z}\right)/2\leq z\leq \left(D_{e}+s_{z}\right)/2$$
(5)$${\left[-\sigma_{w}\frac{\partial \phi }{\partial r}\right]}_{r=R}=-j,\;\mathrm{for}-\frac{D_{e}+s_{z}}{2}\leq z\leq -\left(D_{e}-s_{z}\right)/2,$$
being $D_{e}$ the distance between the central parts of the electrodes, and $s_{z}$ the width of the electrodes in the axial direction. In addition, the current density j in the pipe boundaries, where there are no electrodes, is assumed to be zero. 

The probe conductance G is defined as:(6)$$G=\frac{I}{\delta \phi }=\frac{I}{\langle \phi_{E}\rangle -\langle \phi_{R}\rangle }$$
where *I* denotes the intensity circulating through the electrodes *E* and *R* and δϕ the difference between the average values of the electric potential at the emitter and receiver electrodes. We must note that we are approximating the electrode voltage as the average voltage over the electrode; this approximation has been used by Wang et al. [15], and Tsochatzidis [7]. The reason because of this approximation works in the present study is due to the small size of the two electrodes, which are mounted in the experimental systems that uses conductance probes. $\langle \phi_{E}\rangle$ and $\langle \phi_{R}\rangle$ in Equation (6) are the average values of the potential at the emitter and receiver electrodes, respectively. These average values are computed by means of the expressions:(7)$$\langle \phi_{i}\rangle =\frac{1}{S_{e,i}}\int_{S_{e,i}}^{}\phi \left(r\right)dS$$
where $S_{e,i}$ denotes the area of the i-th electrode and ϕ(r) is the electric potential value at the point defined by the position vector r of the electrode surface. Also, it is assumed, as in the paper by Tsochatzidis et al. [7], that for small electrodes the distribution of current density j is constant over the electrode. With these previous assumptions Equation (6) can be written for a two-ring electrode probe of the same area $A_{e}=A_{E}=A_{R}$ as that described in Figure 1a:(8)$$G=\frac{jA_{e}}{\left(\frac{1}{A_{e}}\int_{A_{E}}^{}\phi \left(R,\;z\right)dS-\frac{1}{A_{e}}\int_{A_{R}}^{}\phi \left(R,z\right)dS\right)}$$
where we assumed that the electrode is at radial position *r* = *R*, and the axial coordinate can vary between the limits of the conductance sensors, being dS=πD dz. Normally the results are expressed in terms of a dimensionless conductance $G^{*}$ which is usually defined as:(9)$$G^{*}=G/\left(\sigma_{w}l\right)$$
where l is the characteristic length of the sensor, and $\sigma_{w}$ the medium conductivity in this case water. The first researchers to obtain an analytical solution of *G* for the two-ring probe were Tsotchatzidis et al. [7]. Some authors prefer to express the results in terms of a relative conductance, that is normally expressed as the ratio of the conductance for a given case and the maximum conductance, that is normally achieved when the pipe is full of liquid. Therefore, we can write:(10)$$G_{rel}^{*}=G^{*}/G_{max}^{*}$$

The expression for the relative conductance can be obtained if one knows the electric potential ϕ(R, z) at the pipe inner surface. This requires solving the Laplace Equation (2), with the boundary conditions of the problem, and then applying Equation (8) to obtain the conductance’s G and $G_{max}$(see Appendix B and Appendix D). The result, for the symmetric case i.e., when the sensor probe is at the center of the cylinder of height H (Figure 1) and with the same amount of water above than below the sensor, is (see Appendix B and Appendix D for more details):(11)$$G_{rel}^{*}=\frac{G^{*}}{G_{max}^{*}}=\frac{\sum_{n=0}^{\infty }\frac{1}{{\left(2n+1\right)}^{3}}\;b_{n}^{2}\frac{I_{0}\left(\gamma_{n}R\right)}{I_{1}\left(\gamma_{n}R\right)}}{\sum_{n=0}^{\infty }\frac{1}{{\left(2n+1\right)}^{3}}\;b_{n}^{2}f\left(\gamma_{n}R_{in},\gamma_{n}R\right)}$$
where the following magnitudes have been defined in Equation (11):(12)$$b_{n}=\cos\left(\gamma_{n}\left(\frac{D_{e}+sz}{2}\right)\right)-\cos\left(\gamma_{n}\left(\frac{D_{e}-sz}{2}\right)\right)\;\mathrm{with}\;\gamma_{n}=\frac{\left(2n+1\right)\pi }{H}$$
where *n* is any integer positive number *n* = 0,1,2,3…and
(13)$$f\left(\gamma_{n}R_{in},\gamma_{n}R\right)=\frac{I_{0}\left(\gamma_{n}R\right)}{I_{1}\left(\gamma_{n}R\right)}\left\{\frac{1+a_{r}\left(\gamma_{n}R_{in}\right)\frac{K_{0}\left(\gamma_{n}R\right)}{I_{0}\left(\gamma_{n}R\right)}}{1-a_{r}\left(\gamma_{n}R_{in}\right)\frac{K_{1}\left(\gamma_{n}R\right)}{I_{1}\left(\gamma_{n}R\right)}}\right\}\;\mathrm{with}\;a_{r}\left(\gamma_{n}R_{in}\right)=\frac{I_{1}\left(\gamma_{n}R_{in}\right)}{K_{1}\left(\gamma_{n}R_{in}\right)}$$
where the functions $I_{0}\left(x\right),\;I_{1}\left(x\right),\;K_{0}\left(x\right)$, and $K_{1}\left(x\right)$ that appear in Equations (11) and (13) with arguments $x=\gamma_{n}R_{in}$ and $x=\gamma_{n}R$ are the modified or hyperbolic Bessel functions of zero, and first orders denoted by the subscripts 0 and 1, respectively. The functions $I_{0}\left(x\right),\;I_{1}\left(x\right)$ are the modified Bessel function of the first class, while $K_{0}\left(x\right)$, and $K_{1}\left(x\right)$ denote the modified Bessel functions of the second class. Notice, that the factor $a_{r}\left(\gamma_{n}R_{in}\right)$, which depends on the internal radius, considers the effect of the internal dielectric cylinder on the dimensionless conductance $G^{*}$ for two-ring conductance electrodes.

In the next case, we assume that the height of liquid above and below the sensor is different, and denoting by $H_{1}$, the height of liquid below the two rings sensor and by $H_{2}$ the height of liquid above the sensor, being $H=H_{1}+H_{2}$. Then on account of the boundary conditions for this problem, the solution for the electric potential can be obtained as explained in Appendix B. Then, substituting the result for the potential in Equation (8) and after some calculus to compute $G^{*}$ and $G_{max}^{*}$ yields for the relative conductance the following result:(14)$$G_{rel}^{*}=\frac{G^{*}}{G_{max}^{*}}=\frac{\sum_{n=1}^{\infty }\frac{1}{n^{3}}a_{n}^{2}\frac{I_{0}\left(\gamma_{n}^{*}R\right)}{I_{1}\left(\gamma_{n}^{*}R\right)}}{\sum_{n=1}^{\infty }\frac{1}{n^{3}}a_{n}^{2}f\left(\gamma_{n}^{*}R_{in},\gamma_{n}^{*}R\right)}$$
where we have defined:(15)$$a_{n}=\sin\left(\gamma_{n}^{*}\left(H_{1}+\frac{D_{e}+sz}{2}\right)\right)-\sin\left(\gamma_{n}^{*}\left(H_{1}+\frac{D_{e}-sz}{2}\right)\right)+\sin\left(\gamma_{n}^{*}\left(H_{1}-\frac{D_{e}+sz}{2}\right)\right)-\sin\left(\gamma_{n}^{*}\left(H_{1}-\frac{D_{e}-sz}{2}\right)\right)$$
With
(16)$$\gamma_{n}^{*}=\frac{n\pi }{H}\;n=1,2,3\ldots$$

The function $f\left(\gamma_{n}^{*}R_{in},\gamma_{n}^{*}R\right)$ is the same defined previously by Equation (13), while $I_{0}\left(x\right),\;I_{1}\left(x\right)$ are the modified Bessel functions of zero and first order respectively and first class. More details can be found in Appendix B.

### 2.2. The Two-Plate Electrodes Conductance Probe

In this case, as mentioned earlier in the introduction, the electrodes can be flush mounted orthogonal to the flow (Figure 2) or parallel to the flow direction as displayed in Figure 3. This kind of electrodes have a good sensitivity for very small fluid thicknesses, but the signal saturates for larger thicknesses. In this section we will study the analytical solution of this problem. The details can be found in Appendix C. 

It is assumed that the current density in the electrodes is constant. This assumption yields good results for small-size electrodes. The first step to solve this problem is to compute the electric potential ϕ(r,θ,z) that now is a function of the three spatial coordinates (r,θ,z). To obtain ϕ(r,θ,z), we must solve the Laplace Equation (2), in cylindrical coordinates by the separation of variables method on account of the boundary conditions. If there is a dielectric inner cylinder of radius $R_{in}$ inside the pipe and the water is located between both cylinders of radius $R_{in}$ and R respectively, then we consider that the current at the interface between the dielectric inner cylinder and the water is zero i.e., ${\left[\sigma_{w}\frac{\partial \phi }{\partial r}\right]}_{r=R_{in}}=0$. In addition, at the inner surface of the pipe and with this arrangement of the electrodes, the boundary conditions at the interface between the electrodes and the fluid are given by:(17)$${\left[-\sigma_{w}\frac{\partial \phi }{\partial r}\right]}_{r=R}=-j=\frac{I}{A_{e}},\;for\;\theta_{1}\leq \theta \leq \theta_{2}\;and\;-s_{z}/2\leq z\leq s_{z}/2$$
and
(18)$${\left[-\sigma_{w}\frac{\partial \phi }{\partial r}\right]}_{r=R}=j=\frac{I}{A_{e}},\;\;for-\theta_{2}\leq \theta \leq -\theta_{1}\;and\;-s_{z}/2\leq z\leq s_{z}/2$$
where $\left[\theta_{1},\theta_{2}\right],$ and $\left[-\theta_{1},\;-\theta_{2}\right]$ are the angular limits of the emitter and receiver electrodes, respectively, and $s_{z}$ the height of the electrodes in the axial direction. In addition, the current density in the pipe boundaries where we have no electrodes is assumed to be zero. Also, we have the boundary condition (3) at the interface between the water and the gas or the dielectric. 

The Laplace equation for the electric potential can be solved as explained in Appendix C, for the different cases. The result is then substituted in Equation (6), to obtain the conductance, and dividing this conductance by the maximum conductance i.e., when the pipe is full of liquid, we get after some simplifications the following result for the relative conductance in the symmetric case, i.e., when the height of water above and below the sensor is the same: (19)$$G_{rel}^{*}=\frac{G^{*}}{G_{max}^{*}}=\frac{C_{1}\sum_{m=1}^{\infty }\frac{a_{m}^{2}}{m^{3}}+\sum_{n=1}^{\infty }\sum_{m=1}^{\infty }\frac{c_{m,n}^{2}}{m^{2}n^{3}}\frac{I_{m}(\gamma_{n}^{\prime }R)}{I_{m}^{\prime }(\gamma_{n}^{\prime }R)}}{C_{1}\sum_{m=1}^{\infty }\frac{a_{m}^{2}}{m^{3}}\frac{\left(1+{\left(\frac{R_{in}}{R}\right)}^{2m}\right)}{\left(1-{\left(\frac{R_{in}}{R}\right)}^{2m}\right)}+\sum_{n=1}^{\infty }\sum_{m=1}^{\infty }\frac{c_{m,n}^{2}}{m^{2}n^{3}}f_{m}(\gamma_{n}^{\prime }R_{in},\gamma_{n}^{\prime }R)}$$
where we have defined:(20)$$C_{1}=\frac{2\pi^{3}s_{z}^{2}R}{H^{3}}\;\;\;\;\mathrm{and}\;\gamma_{n}^{\prime }=\frac{2n\pi }{H}$$

The constant $C_{1}$ depends on the geometric characteristics of the sensor and the pipe. Being $c_{m,n}$ given by the following expression:(21)$$c_{m,n}=a_{m}\left(\theta_{1},\theta_{2}\right)\;sin\left(\frac{\gamma_{n}^{\prime }s_{z}}{2}\right)\;\mathrm{with}\;a_{m}\left(\theta_{1},\theta_{2}\right)=\cos\left(m\theta_{2}\right)-\cos\left(m\theta_{1}\right)$$

The function $f_{m}(\gamma_{n}^{\prime }R_{in},\gamma_{n}^{\prime }R$) that appears in the denominator of Equation (21) is given by the expression:(22)$$f_{m}\left(\gamma_{n}^{\prime }R_{in},\gamma_{n}^{\prime }R\right)=\frac{I_{m}\left(\gamma_{n}^{\prime }R\right)-a_{m,n}\left(\gamma_{n}^{\prime }R_{in}\right)\;K_{m}\left(\gamma_{n}^{\prime }R\right)}{I_{m}^{\prime }\left(\gamma_{n}^{\prime }R\right)-a_{m,n}\left(\gamma_{n}^{\prime }R_{in}\right)\;K_{m}^{\prime }\left(\gamma_{n}^{\prime }R\right)}\;\;\mathrm{with}\;\;a_{m,n}\left(\gamma_{n}^{\prime }R_{in}\right)=\frac{I_{m}^{\prime }\left(\gamma_{n}^{\prime }R_{in}\right)}{K_{m}^{\prime }\left(\gamma_{n}^{\prime }R_{in}\right)}$$
where $I_{m}\left(x\right),\;K_{m}\left(x\right),\;I_{m}^{\prime }\left(x\right),\;K_{m}^{\prime }\left(x\right)$, are first and second class modified Bessel functions of order m and their derivatives. Also, the two-plate conductance probe can be built as displayed in Figure 3, with the two electrodes located in the direction of the flow i.e., the direction of the axis of the cylinder. 

It is important to discuss the physical meaning of Equation (19), the numerator of this equation gives the contribution to the relative conductance, of the conductance of the pipe full of water, while the denominator gives the contribution to $G_{rel}^{*}$ of the conductance of the pipe containing the inner dielectric cylinder. It is important to notice that when the radius $R_{in}$ of the inner cylinder is equal to zero then the first term of the denominator of Equation (19) becomes equal to the first term of the numerator. Also, this same behavior is observed in the second term of the denominator. When $R_{in}=0$, then $a_{m,n}\left(0\right)=0$, and $f_{m}\left(0,\gamma_{n}^{\prime }R\right)=I_{m}\left(\gamma_{n}^{\prime }R\right)/I_{m}^{\prime }\left(\gamma_{n}^{\prime }R\right)$, and therefore the second term of the denominator becomes equal to the second term of the numerator. Therefore, the overall effect is that when $R_{in}=0$, then $G_{rel}^{*}=1$.

In this case the boundary conditions for the potential ϕ(r,θ,z) are according to Figure 3:(23)$${\left[-\sigma_{w}\frac{\partial \phi }{\partial r}\right]}_{r=R}=-j=\frac{I}{A_{e}},for\;-\frac{s_{w}}{2R}\leq \theta \leq \frac{s_{w}}{2R}\;and\;-\frac{D_{e}+s_{z}}{2}\leq z\leq -\frac{\left(D_{e}-s_{z}\right)}{2}$$
and
(24)$${\left[-\sigma_{w}\frac{\partial \phi }{\partial r}\right]}_{r=R}=j=\frac{I}{A_{e}},\;\;\;\;\;for-\frac{s_{w}}{2R}\leq \theta \leq \frac{s_{w}}{2R}\;and\;\frac{D_{e}-s_{z}}{2}\leq z\leq \frac{\left(D_{e}+s_{z}\right)}{2}$$

Also, the current density in the pipe boundaries where we have no electrodes is assumed to be zero as previously. In this case the emitter electrode is the lower one and the receiver electrode is the upper one. The dimensions of each electrode are $s_{w}$ in the azimuthal direction and $s_{z}$ in the axial direction. The distance between the center of the electrodes is $D_{e}$.

The Laplace equation, for the electric potential for the two-plate electrodes flush mounted in the flow direction as displayed in Figure 3, is solved as explained in the second part of Appendix C. The result for the potential is then substituted in Equation (6), to obtain the conductance, and dividing this conductance by the maximum conductance i.e., when the pipe is full of liquid, we obtain after some simplifications the following result for the relative conductance in the symmetric case, i.e., when the height of water above and below the sensor is the same:(25)$$\begin{array}{l} G_{rel}^{*}=\frac{G^{*}}{G_{max}^{*}}\\ =\frac{C_{1}^{\prime }\sum_{n=0}^{\infty }\frac{b_{n}^{2}}{{\left(2n+1\right)}^{3}}\frac{I_{0}(\gamma_{n}R)}{I_{1}(\gamma_{n}R)}+\sum_{n=0}^{\infty }\sum_{m=1}^{\infty }\frac{e_{m,n}^{2}}{{\left(2n+1\right)}^{3}m}\frac{I_{m}(\gamma_{n}R)}{I_{m}^{\prime }(\gamma_{n}R)}}{C_{1}^{\prime }\sum_{n=0}^{\infty }\frac{b_{n}^{2}}{{\left(2n+1\right)}^{3}}f(\gamma_{n}R_{in},\gamma_{n}R)+\sum_{n=0}^{\infty }\sum_{m=1}^{\infty }\frac{e_{m,n}^{2}}{{\left(2n+1\right)}^{3}}f_{m}(\gamma_{n}R_{in},\gamma_{n}R)} \end{array}$$
where we have defined:(26)$$C_{1}^{\prime }=\frac{{\Delta \theta }^{2}}{4},\;\mathrm{and}\;\gamma_{n}=\frac{\left(2n+1\right)\pi }{H},$$
being $\Delta \theta =\frac{s_{w}}{R}$ and
(27)$$e_{m,n}=b_{n}sin\left(\frac{m\Delta \theta }{2}\right)\;\mathrm{with}\;b_{n}=\cos\left(\gamma_{n}\frac{D_{e}+s_{z}}{2}\right)-cos\left(\gamma_{n}\frac{D_{e}-s_{z}}{2}\right)$$

Finally, the functions $f\left(\gamma_{n}R_{in},\gamma_{n}R\right)$ and $f_{m}\left(\gamma_{n}R_{in},\gamma_{n}R\right)$ are the same as those defined by expressions (13) and (22) but with arguments $\gamma_{n}R_{in}\;\mathrm{and}\;\gamma_{n}R$.

In addition, we programmed in MATLAB the previous Equations (19) and (25), deduced in this paper, to obtain the relative conductance for the two-plate conductance probes see the Appendix A. The number of modes we recommended to have good convergence in the results is $n_{max}$ = 95, and $m_{max}=95$, which is equivalent to truncating the number of terms of the series appearing in the numerator and the denominator of these equations to a maximum of 9025 terms per series. Also, we have checked that the contribution of the remaining terms was negligible.

## 3. Comparison of the Analytical Results for the Relative Conductance with the Experimental Results of Different Authors

In this section we check the capability of the previous expressions, developed for different types of conductance probes, to perform predictions of different magnitudes of interest in two-phase flow applications, and liquid fractions.

### 3.1. Comparison of the Analytical Formulas for the Relative Conductance with the Experimental Results for Two-Plate Electrodes

Fossa performed a set of experiments with ring-shape and plate electrodes and compared his results with the available theoretical expressions at that time. For plate electrodes no analytical expressions were available at the time Fossa wrote his paper [8]. Fossa performed four type B tests with two-plate electrodes named B_1_, B_2_, B_3_ and B_4_. For annular flow conditions, the results of test B_3_ were very similar to the previous ones and are not displayed at Figure 4 because they cannot be distinguished in the graphics from the previous ones. Each one of the electrodes had a diameter of 3mm, and were located 9 mm apart in the direction of the pipe axis, so that the distance between the center of the electrodes in Fossa experiments was $D_{e}=9\;\mathrm{mm}+3\;\mathrm{mm}=12\;\mathrm{mm}.$ Fossa [8] measured the relative conductance for annular flow i.e., $G_{rel}=G/G_{max}$ with respect to the conductance of the pipe full of liquid denoted as $G_{max}$. Obviously, this ratio is equal to the dimensionless conductance ratio $G^{*}/G_{max}^{*}$. The ratio values measured by Fossa in tests B_1_, B_2_, B_4_ versus the liquid fraction are displayed in Figure 4 and represented by the blue crosses (x). The pipe internal diameter in these experiments was 14 mm. 

To perform the calculations of the relative conductance for different liquid fractions, we used the same distance between the center of the electrodes as in Fossa’s paper i.e., $D_{e}=12\;\mathrm{mm}$, and we also used the same area for both electrodes i.e., $7.0685\;{\mathrm{mm}}^{2}$. We assume in the calculations displayed in Figure 4 that the electrodes have a square shape being $s_{w}=s_{z}=2.6586\;\mathrm{mm}.$ The square was centered at the same point as the circular electrode. The calculations were performed using Equation (25), and this Equation was programmed in MATLAB. The number of modes used for each calculation was (mmax = 95 and nmax = 95), this involves around $10^{4}$ different terms. As shown in Figure 4, for small liquid fractions the experimental results are very similar to the theoretical ones (inverted green triangles) computed with the formula deduced in this paper for plate probes. In addition, for higher liquid fraction the agreement was also very good. The advantage of using plate electrodes is the good sensitivity to small liquid fraction variations. We have confirmed theoretically this result found experimentally by Fossa.

In addition, we have computed the case of a rectangular electrode centered at the same point as the circular electrodes but with upper and lower sides crossing through the circumference point forming a 45° degrees angle with the x-axis. In this case the height of the electrodes was $s_{z}=3\cos45^{{}^{\circ}}=2.12\;\mathrm{mm}$, and $s_{w}$ is obtained from the condition of maintaining the same area as the circular electrode that yields $s_{w}=3.3\;\mathrm{mm}$. In this last case, the results, the red squares, are a slightly worse for very low liquid fractions ($\alpha_{l}<0.05)$ but slightly better than the previous case for higher liquid fractions ($\alpha_{l}>0.05)$ as displayed also in Figure 4. The number of modes used in both cases was the same one i.e., mmax = 95 and nmax = 95.

It can be shown experimentally that there is not a significant difference using plate electrodes if they are mounted in the axial or the azimuthal direction if the distance between the center of the electrodes and their geometry is the same when the radius of the pipe is large enough. 

### 3.2. Comparison of the Changes of the Relative Conductance with the Liquid Fraction for Homogeneous Bubbly Flow with the Experimental Results of Fossa

To obtain the fraction of liquid from measurements of the relative conductance performed with flush-mounted two-ring electrode probes in two-phase flow homogeneous mixtures, we use the expression obtained in Appendix D (Equation (A40)) for the relative conductance of a two-ring electrode. Assuming a homogeneous two-phase mixture as, for instance, homogeneous bubbly flow, one obtains:(28)$$G_{rel}=\frac{G_{\alpha }}{G_{l}}=\frac{\sigma_{eff}}{\sigma_{l}},$$
where $G_{\alpha }$ is the conductance for the two-phase homogeneous mixture with void fraction α, and $G_{l}$ the conductance when the pipe is full of liquid, $\sigma_{eff}$ is the effective electrical conductivity for the two-phase mixture and $\sigma_{l}$ is the liquid electrical conductivity. The effective medium theory (EMT) replaces the heterogeneous media properties by a homogeneous or effective medium having the same response to the excitations. Two assumptions can be made to deduce the effective expression for the conductivity, the first being known as the “non-consistent” hypothesis is to assume that the host phase is one of the phases of the mixture. In this case if one considers that the host phase is one of the phases of the mixture and that all the inclusions except the host have spherical geometry, then assuming that the liquid is the carrying phase it is found that the expression that gives the effective conductivity is [16]:(29)$$\frac{\sigma_{eff}-\sigma_{l}}{\sigma_{eff}+2\sigma_{l}}=\underset{i=l,g}{\sum }\alpha_{i}\frac{\sigma_{i}-\sigma_{l}}{\sigma_{i}+2\sigma_{l}}$$
where $\alpha_{l}$, and $\alpha_{g}$ are the liquid and gas volumetric fractions, respectively. Because the conductivity of the gas is negligible compared with the liquid phase conductivity, then we can assume in Equation (29) that $\sigma_{g}≅0$, and one deduces from Equation (29) Maxwell equation for the effective conductivity [17]:(30)$$\frac{\sigma_{eff}}{\sigma_{l}}=\frac{2\alpha_{l}}{3-\alpha_{l}}$$

The other assumption is to consider that the hosting substance has an effective conductivity. This is equivalent to consider that the bubbles are embedded in an effective medium with conductivity $\sigma_{eff}$ which is the same for all the bubbles [16,18,19]. This assumption yields the self-consistent EMT result, and the expression for this approximation is obtained taking in Equation (29), $\sigma_{eff}=\sigma_{l}$, which yields when the diluted entities are spherical gas bubbles:(31)$$\alpha_{g}\frac{\sigma_{g}-\sigma_{eff}}{\sigma_{g}+2\sigma_{eff}}+\alpha_{l}\frac{\sigma_{l}-\sigma_{eff}}{\sigma_{l}+2\sigma_{eff}}=0,$$
where $\sigma_{l},\;\sigma_{g},\;\sigma_{eff}$ are the electric conductivities, of the liquid, the gas and the two-phase mixture. Then, Equation (31) when we assume that the gas conductivity is zero simplifies to the following expression for $\alpha_{l}>1/3$:(32)$$\frac{\sigma_{eff}}{\sigma_{l}}=1.5\;\alpha_{l}-0.5$$

Other expressions commonly used are: the Begovich and Watson equation [6], not shown here because does not predicts well the experimental data of Fossa for bubbly flow, and the Bruggeman expression that is given by [20]:(33)$$\frac{\sigma_{eff}}{\sigma_{l}}=\alpha_{l}^{3/2}$$

Fossa performed a set of experiments using a two-electrodes ring conductance probe, with a pipe diameter of 70 mm, width 6 mm and two distances between the centers of the electrodes De=30 mm, and 20 mm. The measurements were performed inside a cylindrical pipe of 48 cm height. We display in Figure 5 the comparison with the Fossa results for the De=30 mm test using bubbly flow conditions. It is observed that for liquid fractions above 0.85 all the expressions predict very well the experimental data. Below $\alpha_{l}=0.85$ the expression that better predicts the experimental data is EMT theory using the non-consistent hypothesis which yields the Maxwell equation [17], and the worst is the EMT with the self-consistent hypothesis (EMT-SC).

Recently Wang et al. developed a new empirical model, valid for churn flow and slug flow, which relates the water holdup with the relative conductance of the two-phase mixture [21].

### 3.3. Comparison with Coney Experiments

In this section we discuss the results obtained using the expressions deduced in Section 2.2 for the conductance ratio $G/G_{max},$ using different geometric characteristics of the conductance probe, and comparing them with the experimental results obtained by Coney [10]. This author made the conductance probe of two electrodes parallel to the axis of the pipe, being the distance between the electrode edges denoted by 2a, with a approximately 1 mm, see Figure 6a. The electrodes were made of stainless steel, and Coney used two kinds of electrode for the probe the first one denoted in this paper as Coney-short had an overall length along the axis of $l_{0}=8.3\;\mathrm{mm}$, while the electrode denoted as Coney-long had an overall length of $l_{0}=26.7\;\mathrm{mm}$. The receiver electrode for both probes was divided in three segments, with the central one having a length of approximately 2.7 mm for the short probe and 2.88 mm for the long probe, while each one of the two outer segments in the short probe had a length of 2.7 mm while in the long probe this length was 11.9 mm. The individual segments of the receiver electrode were insulated with paper impregnated with Perspex cement producing a separation of 0.1 mm between segments, as displayed at Figure 6b. The electrodes had a width of 2 mm along the pipe circumference in the short probe and 2.15 mm in the long probe which determined the angular position of the electrodes. The internal diameter of the pipe was 2.54 cm, and the electrode surfaces were machined to have a curved surface of 2.54 cm diameter.

Coney used a Perspex disc with its surface machined into steps of different radius so different film thickness can be created by moving the disc. Then, Coney measured $G/G_{max}$ in terms of the water thickness δ and expressed these measurements in terms of δ/a, for both probes being 2a the separation between the two electrodes in Coney experiments [10]. Because the half distance was approximately a = 1 mm, this was equivalent to expressing the relative conductance of the probe in terms of the water thickness in mm. Figure 7 displays the results obtained with Equation (19) for the relative conductance of the probe in Coney experiments using the total length of the electrodes and their geometric characteristics. The theoretical results match the experimental ones very well when using the total length of the electrodes i.e., $l_{0}=26.7\;\mathrm{mm}$ for the longest one and $l_{0}=8.3\;\mathrm{mm}$ for the shortest.

However, if the separation of the electrodes is not small compared with their length as happens when we consider two electrodes with length equal to the central segment length l=2.88 mm, then the discrepancy with the experimental data slightly increases as displayed in Figure 8. The theoretical results obtained in this case with Equation (19) are given by the upper curve of Figure 8. In this case the calculations were performed assuming that both electrodes have a length of 2.88 mm, and the rest of characteristics remain the same as in the experiment. However, Coney found experimentally that when one of the electrodes is segmented into three parts by very small (0.1 mm) insulating strings, then the outer segments act as a guard ring. Consequently, the current to the central segment is close to the theoretical prediction of the Coney formula without segmenting assuming very long electrodes with l≫a, which allows to neglect edge electric field effects at the end of the electrodes. Obviously in this case because the separation among the electrodes is 2a = 2 mm, edge effects are important and cannot be neglected. Therefore, if we use the total length of the electrode we approach with Equation (19) the experimental results, obviously this does not happen with Coney formula because it is 2D, and he assumes a very long electrode in the axial direction. For very long electrodes, the Coney formula gives the results displayed in the lower curve of Figure 8. 

There remains a question to be investigated: if Equation (19) is correct then if we make both electrodes very long, for instance 60 mm, the edge effects will be small and our results using Equation (19) should be close to the results obtained with Coney equation. We have represented in Figure 9 the results obtained using a probe with an electrode length equal to 60 mm, separation between electrodes of 2 mm, and with electrodes having a width of 2.15 mm. Both the Coney equation, which assume an infinite length of the electrodes, and the 3D formula deduced in this paper yields results which are very close that confirm the validity of Equation (19). 

### 3.4. Comparison with Ko et al. Data and Lee et al. Data

In this section we compare the results of the analytical formulas deduced previously and programmed in MATLAB with the experimental results of Ko et al. [12], and Lee et al. [13]. These authors performed several static experiments with annular flow. The goal was to verify experimentally the sensor design that they obtained with the program COMSOL, which uses the finite element method (FEM) to solve the Laplace equation with the appropriate boundary conditions imposed by the electrodes at the inner surface of the pipe and the boundary conditions of zero current at the rest of the surfaces. They fabricated a conductance sensor mounted as displayed in Figure 10, with an inner diameter of 40 mm and three electrodes A, B, C flush located along the inner circumferential direction of the tube. Two of the electrodes A and B, have the same length and span of 2.54 radians each, and the third one, the C span, has 0.3 radians with two insulator sections of 0.2 radians at both sides; in the bottom there was a section of insulator spanning 0.5 radians. 

Then, Ko et al. [12] performed a set of measurements of the potential drop between the electrodes A and B, or alternatively A and C. The annular flow was achieved inserting acrylic rods of different diameters in the pipe containing the sensor. They measured the conductance G, between the electrodes A and B, for different radius of the rods, and also the conductance $G_{l}$ for the pipe full of liquid, obtaining the ratio Grel=GGl or relative conductance. The results obtained by these authors are displayed in the upper line of Figure 11a, and the relative conductance shows a small difference with the linear-conductance response $G_{rel}=\alpha_{l}=1-\alpha$ around +0.06 on average or +6% when expressed in percent. We computed using Equation (19) and Equation (A39) the relative conductance for this same case, and we obtained the results displayed as the lower line (blue) displayed at Figure 11a, which deviates −1% from the linear-conductance response. 

The next issue was to compare the results obtained by Lee et al. [13] with the analytical results and the linear-conductance response. The experiments performed by Lee et al. [13] were similar to those performed by Ko [12] et al., the geometrical configuration of the sensor was the same but in this case the inner diameter of the sensor was 45 mm instead of 40 mm as in the previous case. The results obtained by Lee et al. are displayed in the upper line of Figure 11b (blue), and the results show a difference of +5.7% with respect to the linear-conductance response denoted as the Begovich–Watson line (green), which is the middle line. Finally the results obtained using Equations (19) and (C7), are displayed in the lower line of Figure 11b, and show a difference with the linear-conductance response of −1%, and for the void fractions that are below 0.3 this difference is practically zero.

## 4. Discussion of Results, their Interpretation and Implications, Future Trends, and Final Conclusions

In this paper we have deduced the analytical expressions for the relative conductance and the potential difference for two plate-electrode conductance probes in two configurations: the first one is when the electrodes are flush mounted in the flow direction i.e., along the z direction of the pipe axis as displayed in Figure 3, and the second one is when the electrodes are flush mounted along the circumferential direction of the pipe as displayed in Figure 2a,b. All these expressions are fully 3D and have been deduced solving the 3D Laplace equation with a proper boundary condition, as shown in the Appendix C and Appendix D, assuming that the current density is constant over each electrode and that the frequency is high enough to neglect capacitive effects. Also, we have assumed an average potential over each electrode, that have been obtained using Equation (7), so the potential difference between the electrodes has been calculated as the difference between the averaged electric potential over each separate electrode. Previously Coney [10] obtained a very well-known expression for a conductance probe consisting in two flush-mounted parallel electrodes of unequal widths and infinite length and separated by an insulator. The expression deduced by Coney was checked experimentally by many authors such as Fossa [8], Tsochatzidis et al. [7], and Coney himself [10]. Because Coney used parallel finite length electrodes flush-mounted inside a pipe, his experimental results approach the value deduced by himself for the relative conductance in terms of the liquid fraction but never attains the analytical results. We have checked this in this paper, because the analytical expression is fully 3D as a result of the analytical Equation (19) obtained in this paper, that the relative conductance in terms of the liquid fraction exactly matches the experimental results obtained by Coney with smaller electrodes as displayed in Figure 7. In addition, we have found that Equation (19) for the relative conductance between two plate parallel electrodes approach the result of the expression obtained by Coney as the electrode lengths becomes large, as can be observed in Figure 9. Also, it is deduced using the new expression, as displayed in Figure 8, that when the height of the electrodes becomes smaller the relative conductance attains faster the saturation value and the slope of the curve increases. This means that the measurements are more sensitive to small variations of liquid thickness. However, the liquid film thicknesses we can measure are smaller.

In addition, Fossa performed a set of experiments with two-plate electrodes, with the electrodes located along the flow direction, and at 12 mm distance between the electrode centers. For this case Fossa measured the relative conductance for different liquid fractions and obtained the results displayed at Figure 4 (blue x), with electrodes that had a circular shape. We have made the calculations with square electrodes of the same area using Equation (25), and the results are represented by the green inverted triangles that agree with the experimental data for liquid fractions below 0.1, and show a very small difference above 0.1. Then we performed the calculations assuming that the electrodes have the same area than the circular electrodes and were centered at the same point as displayed in Figure 4b, but the upper and lower sides of each electrode crossed through the circular electrode point that formed a 45° angle with the x-axis; in this case the experimental results matched exactly the analytical ones obtained with Equation (25). The total number of modes used to perform the calculations was $10^{4}$, and the solution is obtained in a few seconds with a PC, having programmed the equation in MATLAB. Therefore, we conclude that the shape of the electrodes (circular, square or rectangle) can have some influence on the results but this influence is small if the area and the location of the electrode centers are the same and the relation of the length to the height of the electrode dimensions are within the following limits ($1\leq s_{w}/s_{z}\leq 1.6$).

Another question to be discussed is the influence of the number of modes in the results, and what are the optimal values of $n_{max}$ and $m_{max}$. To answer this question at Table 1 we compare the results obtained for the relative conductance of the case displayed as the upper curve of Figure 8, using $n_{max}=95$, and $m_{max}=95$, with the same case performed using $n_{max}=100$, and $m_{max}=100$. The difference in the number of terms contributing to the result in the numerator and the denominator of Equation (19) for both cases is 980. It is observed that the influence of adding these extra terms on the results is always smaller than 0.0012.

We have seen that diminishing the length of the electrodes in the Coney experiments, as displayed in Figure 8, increases the relative conductance $G/G_{max}$ versus delta, for a fixed distance 2a between the electrodes. In Figure 8 the distance between the electrodes was fixed at 2 mm, and the electrode length varied from the smaller one (2.88 mm) for the upper curve “$G/G_{max}$ versus delta”, to the largest one of the lower curve. So, it is concluded that when the length of the electrodes or their guard electrode lengths diminishes, then the relative conductance increases faster especially for lower values of delta. This means that the two-plate detector attains faster the saturation for smaller electrode sizes in the axial direction. What happens when we maintain fixed the length of the electrodes and we increase the distance between the electrodes? The response to this question is displayed in Figure 12. We have performed the calculation of the relative conductance for two parallel electrodes with $s_{z}=2.88\;\mathrm{mm}$, a pipe diameter of 5.08 cm, and $s_{w}=2.15\;\mathrm{mm}$. Then we changed the distance 2a between the electrodes, and performed the following cases a=0.6 mm, a=0.8 mm, a=1 mm, a=1.2 mm. The results for the relative conductance are displayed in Figure 12. The results of this figure tell us that diminishing the distance between the electrodes for fixed values of their sizes increases the slope of the curve $\frac{G}{Gmax}$ versus delta. The quantitative effect on the slope is small and, therefore, the effect on the saturation is small.

In addition, to the previous comparisons with two plate electrodes, we performed a comparison with the sensor designs of Ko et al. [12], and Lee et al. [13], that consisted in two plate electrodes orthogonal to the flow, and spanning 2.54 radians each electrode and with different separation at both sides 0.7 radians, and 0.5 radians, respectively. The sensor designs of both authors were very similar, the only difference being the internal radius, which was slightly different. These designs were performed to obtain a response as close as possible to the linear conductance response, where the linear response was set to $G_{rel}=G_{\alpha }/G_{l}=1-\alpha$, being $G_{\alpha }$ the conductance for a void fraction α, and $G_{l}$ the conductance when the pipe was full of liquid. The conductance for a void fraction α was achieved by these authors inserting small rods of an acrylic non-conducting material, and then they measured the conductance ratio for different radius of the acrylic inner cylinder. Then they obtained the results displayed in Figure 11a,b for the annular flow, which are slightly above (6%) the linear behavior, denoted as Begovich–Watson, as can be observed in Figure 11b. The results obtained using Equation (19), deduced in this paper are slightly below (1%) the linear behavior curve, and are also displayed in Figure 11a,b. At this point, it is convenient to remark that the electrical signals of a sensor based on the electric-impedance between two electrodes depends not only on the fraction of liquid or the void fraction in the sensor region but in addition of the liquid distribution inside the pipe which in turn depends on the flow regime (bubbly, slug, annular…) that we have in that region as will be discussed below.

The next issue to be discussed is the measurement of the liquid fraction ($\alpha_{l}$) and the void fraction ($\alpha =1-\alpha_{l})$ from measurements of the relative conductance performed with flush-mounted two-ring electrode probes in two-phase flow homogeneous mixtures. In this case assuming a homogeneous two-phase mixture as for instance homogeneous bubbly flow, one obtains as deduced in Appendix D, Equation (A40), which shows that the ratio of the conductance of the two-phase mixture to the conductance of the pipe full of water is equal to the ratio of the conductivities of the mixture and the water $G_{rel}=G_{\alpha }/G_{w}=\sigma_{eff}/\sigma_{w}$. $\sigma_{eff}$ is the effective conductivity of the two-phase mixture, which depends not only on the void fraction but also on the two-phase distribution. For homogeneous bubbly flow distribution, as in the experiments performed by Fossa [8], the best prediction of the relative conductance in term of the liquid fraction is obtained using the Maxwell formula. However, Yang and Kim [22] measured the relative resistance $R_{water}/R_{2\phi -mixture}$, using different types of conductance probes, for type II probes, which are two-electrode probes formed by two electrodes A, and B spanning less than pi radians of each electrode, with a radius of 6 cm and a height of 6 cm. Moreover, the frequency used in their experiments was 100 kHz so the capacitive part of the impedance was small and we can write $R_{water}/R_{2\phi -mixture}≅G_{\alpha }/G_{w}=\sigma_{eff}/\sigma_{w}$. In the experiments of Yang and Kim [22], the air–water mixture was not homogeneous as in Fossa experiments, as it is deduced observing the figures of the void distribution of their paper [22]. The results obtained by Yang and Kim with probe II for the relative resistance are displayed in Figure 13 of this paper. It is observed in this case that the best prediction of the experimental data is obtained with the self-consistent EMT theory, in this approach of the effective medium theory, one assumes that the bubbles are embedded in an effective medium with conductivity $\sigma_{eff}$, which is the same for all the bubbles as explained in Section 3.2. These results are different from those obtained with the Fossa experiments where the Maxwell formula gives the best predictions. We must consider that for the case displayed in Figure 13, the bubbly flow is not homogeneous, contrary to the case displayed in Figure 5, where it is homogeneous. Therefore, for Yang and Kim experiments the self-consisting EMT theory gives the best predictions for the non-homogeneous bubbly flow that the non-consistent EMT (Maxwell equation). This result is coherent because of the Maxwell equation for the mixture conductivity is based on the hypothesis of homogeneity.

To finish we conclude that the formulas obtained in this paper for the two electrode conductance probes in cylindrical geometry predict the relative conductance very well in terms of the liquid fraction for different set of experiments and different sizes and geometries of the sensors, and the results can be obtained in a few seconds using $10^{4}$ modes.

We have seen that when using the sensors for holdup applications to predict the average void fraction in a region, the results depend not only on the void fraction but also on the two-phase distribution. Therefore, in the simple case of bubbly flow as in Fossa experiments, which uses a homogeneous flow, the experimental results show that the Maxwell equation is the best suited to predict the void fraction. However, for bubbly but not homogeneous flow, the self-consistent EMT equation is that which most approaches the experimental data obtained by Yang and Kim [22]. Future research directions could study the influence of wall peak and core peak void fraction distribution for bubbly flow on the conductance or resistance ratios. Also, an interesting question that is now becoming relevant is if we can get the liquid fraction from relative conductance measurements using conductance probes for slug and churn turbulent flows i.e., for low water holdup structures. This question has been addressed recently by Wang et al. [21,23] and Yang et al. [24]. These authors arrive from the experimental data, by a fitting procedure, to the result that this relation is given for this type of flow by:(34)$$\alpha_{l}={\left(G_{rel}\right)}^{n}\;with\;n=1.5016\;\mathrm{or}\;\alpha_{l}≅{\left(G_{rel}\right)}^{3/2}$$

For more general flow Yang et al. [24] propose a general expression that it is a weighting average of Maxwell expression for bubbly flow and the Wang et al. [21] equation for slug flow and they write:(35)$$\alpha_{l}=w_{h}\frac{3G_{rel}}{2+G_{rel}}+w_{l}{\left(G_{rel}\right)}^{1.5016}$$
where, according to these authors, $w_{h}$ is the weight of the high-water holdup structures, and $w_{l}$ is the weight of the low water holdup structures. These authors determine these weights experimentally counting the number of sampling points $N_{h}$ of signals of high-water holdup structures, and the number of sampling points of signals $N_{l}$ of low water holdup structures. Then $w_{l}=N_{l}/N$, and $w_{h}=N_{h}/N$. N, is the total number of sampling points.

## Figures and Tables

**Figure 1 sensors-20-07042-f001:**
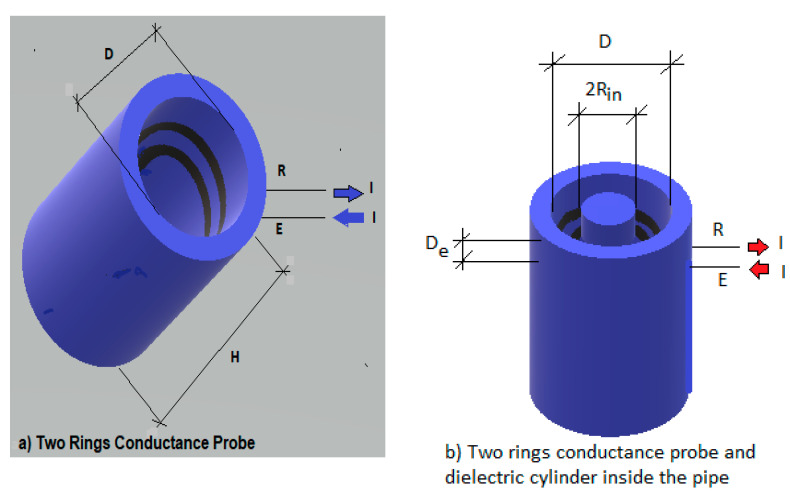
(**a**) Two rings conductance probe configuration with the pipe full of water; (**b**) two-ring conductance probe with an inner dielectric cylinder inside the pipe.

**Figure 2 sensors-20-07042-f002:**
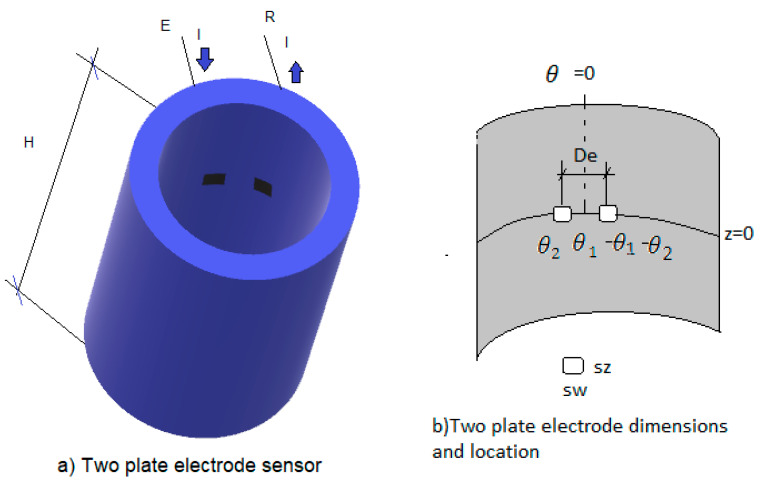
(**a**) Conductance probe of two plate electrode configuration with the pipe full of water; (**b**) notation and location of the electrodes in cylindrical coordinates.

**Figure 3 sensors-20-07042-f003:**
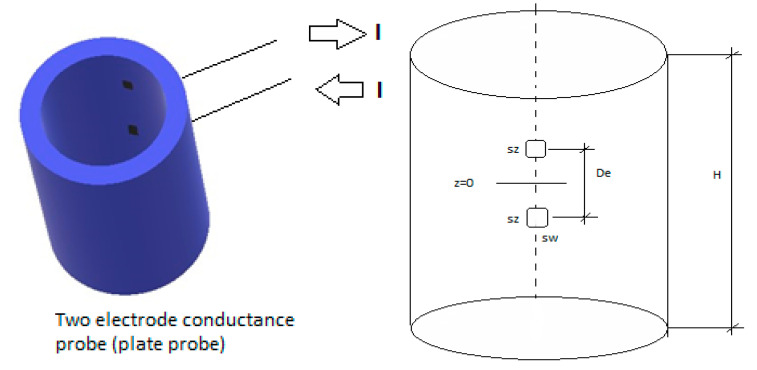
Two-plate electrode probe along the direction of the pipe axis.

**Figure 4 sensors-20-07042-f004:**
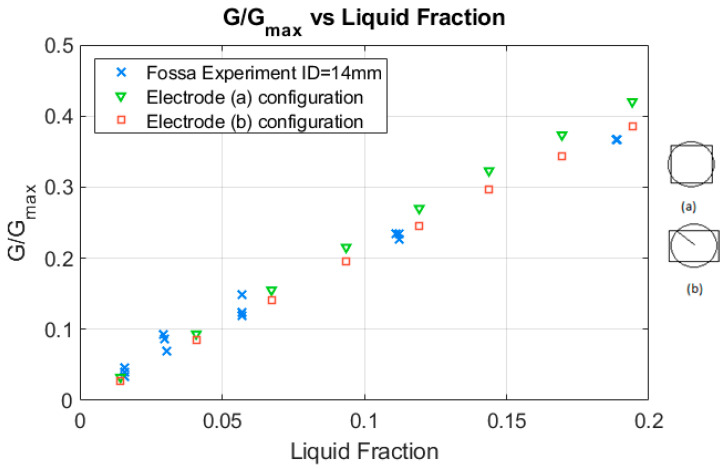
Theoretical versus experimental (blue x) results for the experiments performed by Fossa [8] with a sensor with two-plate electrodes inside a tube with 14mm ID. For the green inverted triangles, the calculations were performed with a square electrode of the same area (**a**). For the red squares, the calculations were performed with the electrode displayed in figure (**b**), of equal surface area than the circular electrode.

**Figure 5 sensors-20-07042-f005:**
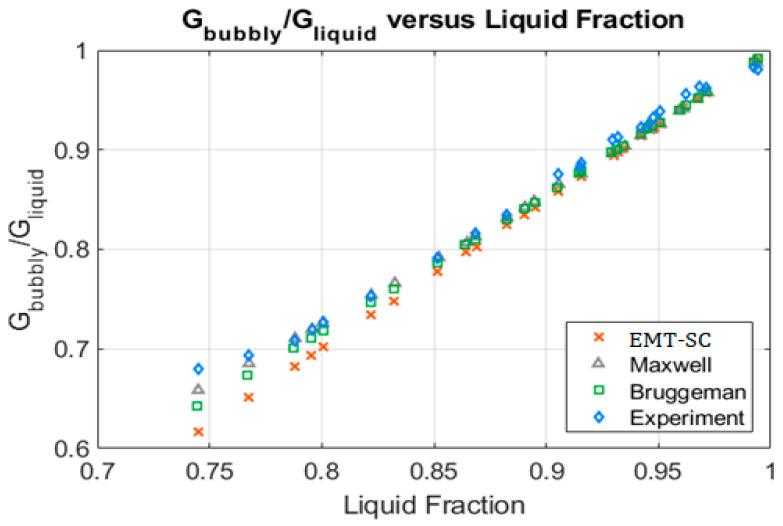
G(Bubbly)/G(liquid) versus liquid fraction for experimental data (blue rhomboids) Bruggeman formula (green squares), Maxwell formula or effective medium theory (EMT) non-consistent (grey triangles), EMT self-consistent (orange X).

**Figure 6 sensors-20-07042-f006:**
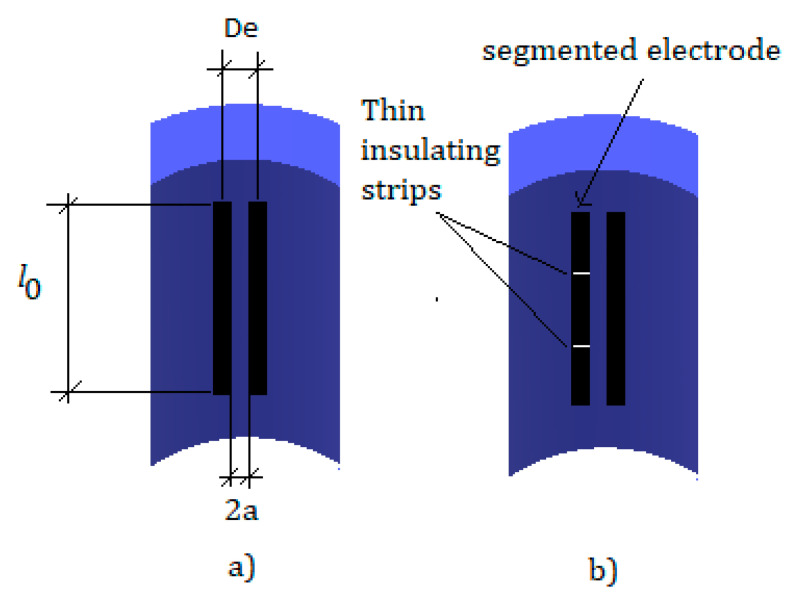
(**a**) Conductance probe with two parallel rectangular electrodes, (**b**) similar probe with one of the electrodes segmented as designed by Coney [10].

**Figure 7 sensors-20-07042-f007:**
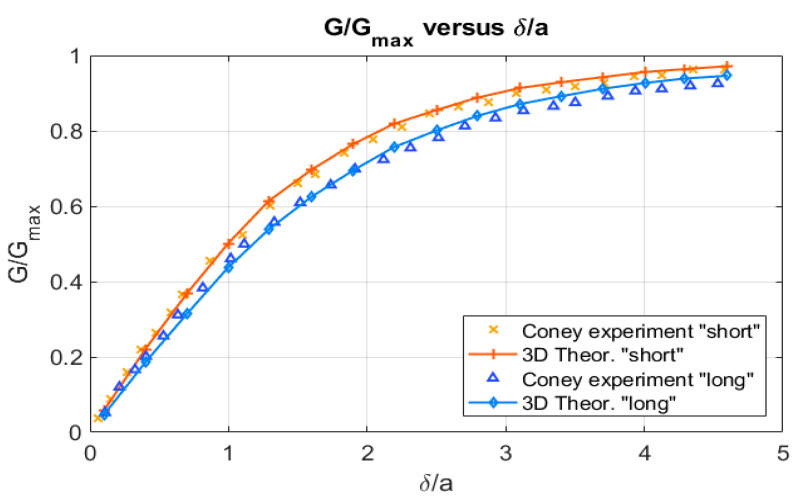
$G/G_{max}$ versus δ/a, for the experiments performed by Coney with the conductance probe, with only one of the electrodes segmented, and long and short electrodes and the results of the calculations with the formulas of Section 2.2.

**Figure 8 sensors-20-07042-f008:**
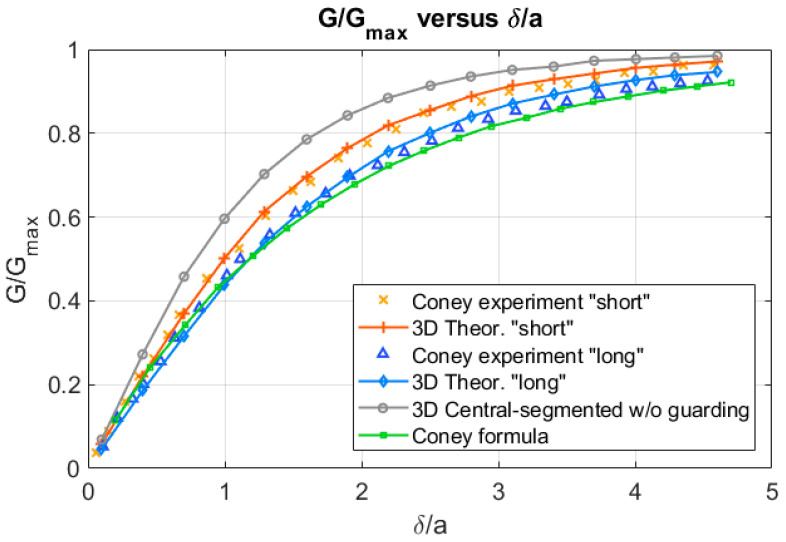
$G/G_{max}$ versus δ/a, for different cases explained in the text. The upper curve grey line has been computed with Equation (19) for G/Gmax and l=2.88 mm, the lower green line has been computed with the Coney formula which assumes l≫a.

**Figure 9 sensors-20-07042-f009:**
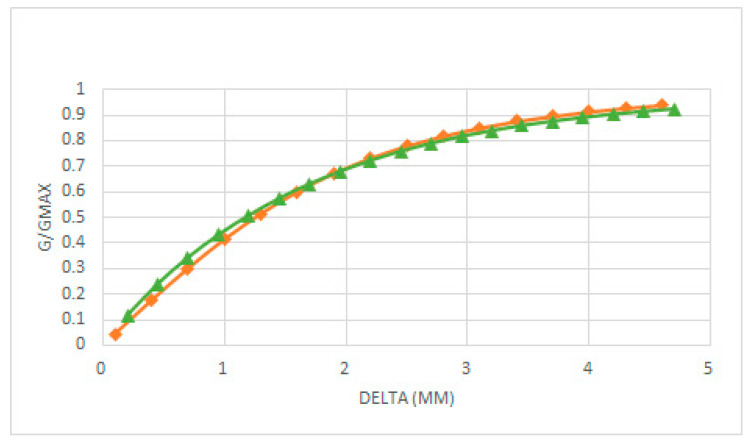
G/Gmax versus δ computed with Coney formula valid for l≫a with the characteristics of the previous experiments (green-triangles) and the 3D formula obtained in this paper’s Equation (19) using large l=60 mm≫a=1 mm (rhomboids).

**Figure 10 sensors-20-07042-f010:**
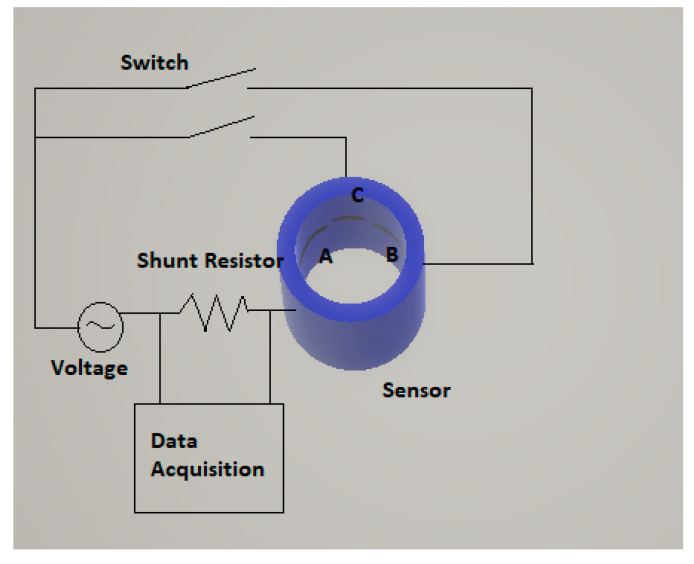
Schematic diagram of the sensor system used by Ko et al. [12]. The sensor has three electrodes (**A**–**C**) and the measurements can be performed using electrodes (**A**,**B**) or (**A**,**C**).

**Figure 11 sensors-20-07042-f011:**
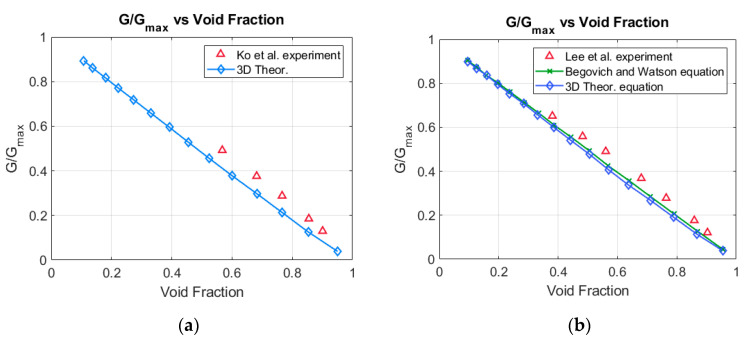
(**a**) G/Gmax versus the void fraction for the experiments performed by Ko et al. [12], displayed in the upper line and the calculations performed with Equation (19), displayed in the lower line. (**b**) G/Gmax versus the void fraction for the experiments performed by Lee et al. [13], upper-line and the calculations performed with Equation (19), lower line, the middle line are the results obtained using the Begovich and Watson equation [6].

**Figure 12 sensors-20-07042-f012:**
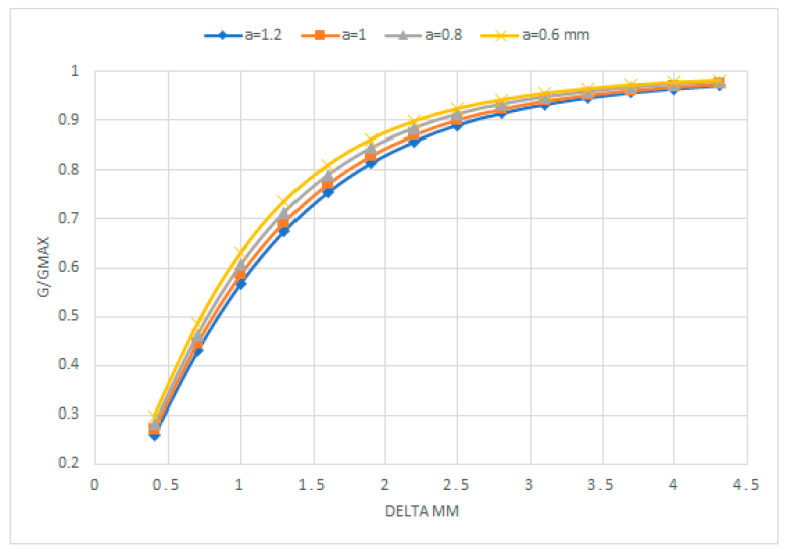
$\frac{G}{G_{max}}$ versus δ, for different distances between the electrodes. 2a=1.2 mm, 2a=1.6 mm, 2a=2 mm, and 2a=2.4 mm.

**Figure 13 sensors-20-07042-f013:**
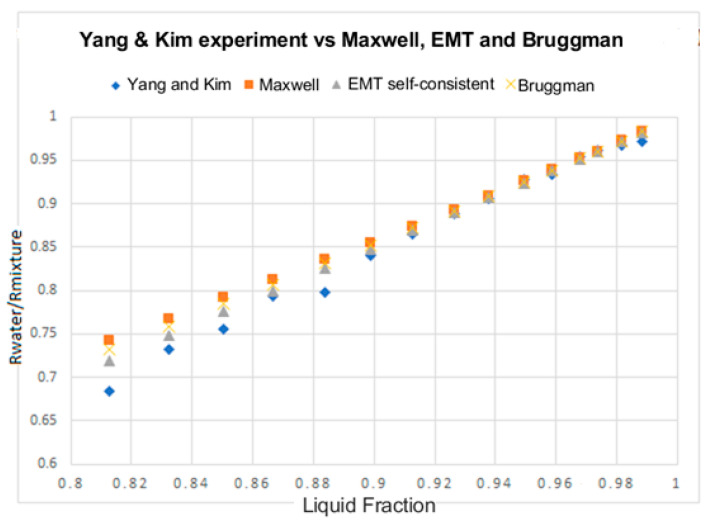
Non-dimensional resistance ratio (RwRmixture), versus liquid fraction for probe II, of Yang and Kim experiments [21], and comparison with Maxwell results (squares), EMT self- consistent (triangles), and Bruggeman (Saint Andrew crosses).

**Table 1 sensors-20-07042-t001:** Relative conductance $G/G_{max}$, computed with $n_{max}=95$, and $m_{max}=95$, and wit $n_{max}=100$, and $m_{max}=100$ for different values of δ/a, for the case of curve grey line of Figure 8.

δ/a	$G/G_{max}\;95\times 95$	$G/G_{max}\;100\times 100$
0.4	0.2694	0.2703
0.7	0.4442	0.4454
1.0	0.5850	0.5861
1.3	0.6923	0.6933
1.6	0.7715	0.7723
1.9	0.8291	0.8298
2.2	0.8708	0.8714
2.5	0.9012	0.9016
2.8	0.9234	0.9237
3.1	0.9399	0.9402
3.4	0.9522	0.9525
3.7	0.9616	0.9618
4.0	0.9688	0.9690
4.3	0.9745	0.9746

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
