# Peer review of "Analysis of Conductance Probes for Two-Phase Flow and Holdup Applications"

_sensors, 2020, doi:10.3390/s20247042_

Round 1
Reviewer 1 Report
The manuscript is an interesting study for analysing the effect of conductance probes in two-phase annular flows. The study is of interest to the readers and the community of multiphase systems. I recommend the manuscript for publication considering the following comments:
1- It is not clear what is the main novelty of the present study with respect to other studies in literature. This will improve the readability of the manuscript and is of interest to readers. So the authors should clearly explain the novelty of the manuscript towards the end of the introduction section. The authors can also indicate it in the conclusion section.
2- The effect of electrohydrodynamics has a broad range of applications and implemented by a myriad of numerical and experimental methods. I suggest the authors to improve the introduction considering a wider range of numerical models. Some examples are enclosed as suggestions:
https://doi.org/10.1016/j.ijheatfluidflow.2017.01.002
https://doi.org/10.1115/1.4047593
https://doi.org/10.1016/j.ces.2020.115819
3- Some of the equations, e.g. eq. 13, 19, 25 should be modified. It is really unclear to follow them.
4- Please provide a table of nomenclature and make sure that all variables are defined.
5- One major comment is that the authors mainly tried to verify their results with several experiments and analytical results, etc. This is understandable that good validation is necessary, but they should provide original outcomes with clear discussion and explain what they have added. It its current format, the weight of the validation is much larger that the weight of original discussion and results.
Author Response
Please see the attachment file for responses and comments

Reviewer 2 Report
Review of Analysis of Conductance Probes for Two-Phase Flow and Holdup Applications The topic is relevant and challenging. The paper is well written and organized, and gives relevant results. The paper is worth publishing in Sensor after the authors address the following comments: 1-The English should be checked, there are some typos
2-section 3.2 : Analytical model is based on assumptions of homogeneity and spherically of the bubble, but it seems also that assumption should be made on the size of the bubbles. Is the model valid for very large bubbles?
3-line 313 : it should be noted that conductivity of gas is negligible compared to the son of the liquid phase, saying it is almost zeros is not meaningful from a physical point of view.
4-section3.3: it is very difficult to understand the geometry of the experiment a detailed scheme would be of great help.
5-section3.3: Figure 7 what is exactly delta/a, how does it relate to void fraction?
6-in the model development it is not clear what is the analytical part and what is the numerical part, and is done the numerical part. Could you clarify
Author Response
Please see the attachment for response to comments and corrections

Reviewer 3 Report
The paper introduced a method using the electric potential and the relative conductance to solve analytically the generalized Laplace equation in 3D with the boundary conditions, studying the different types of conductance probes analytically, comparing these analytical results with numerical results and experiments of different authors. This paper has certain significance for the accurate measurement of liquid film with conductance method. Though the work presented in this paper has the potential to be useful, it requires a major rewrite before it can be accepted.
1. The experimental data within the liquid holdup range given in Fig. 4 are relatively small, which cannot effectively support the author's conclusion. It is suggested that the author provide more relevant data.
2. The division of low liquid holdup and high liquid holdup is not clear and many qualitative conclusions are given. I suggest using quantitative expressions.
3. Why the number of modes (m and n) was 95? The author should give relevant evidence.
4. The conclusion “Below ??=0.85 the expression that better predict the experimental data is EMT theory using the non-consistent hypothesis which yield Maxwell equation” is not consistent with the results shown in Figure 5.
5. There are many language problems in this paper, such as “the same area also than the circular electrode in Fig.5”, “Equation (31) when we assume that the gas conductivity is zero reduces to the following expression for ??>1/3” etc.
Author Response
Please see the attachment for responses and comments

Round 2
Reviewer 3 Report
The revised article has answered most of the queries raised by the Reviewers. However, the conclusion “Below ??=0.85 the expression that better predict the experimental data is EMT theory using the non-consistent hypothesis which yield Maxwell equation” is not consistent with the Fig.5 description, the Fig.5 showed that the predictive result of EFM deviate more from the experimental results, We look forward to a full and reasonable explanation from the author.
The article needs some more editorial corrections to bring more clarity to the readers.
